# A Novel Camera-Based Measurement System for Roughness Determination of Concrete Surfaces

**DOI:** 10.3390/ma14010158

**Published:** 2020-12-31

**Authors:** Barış Özcan, Raimund Schwermann, Jörg Blankenbach

**Affiliations:** Geodetic Institute and Chair for Computing in Civil Engineering & Geo Information Systems, RWTH Aachen University, Mies-van-der-Rohe-Str. 1, 52074 Aachen, Germany; schwermann@gia.rwth-aachen.de (R.S.); blankenbach@gia.rwth-aachen.de (J.B.)

**Keywords:** concrete surface roughness, 3D reconstruction, digital photogrammetry, non-destructive testing

## Abstract

Determining the roughness of technical surfaces is an important task in many engineering disciplines. In civil engineering, for instance, the repair and reinforcement of building component parts (such as concrete structures) requires a certain surface roughness in order to ensure the bond between a coating material and base concrete. The sand patch method is so far the state-of-the-art for the roughness measurement of concrete structures. Although the method is easy to perform, it suffers from considerable drawbacks. Consequently, more sophisticated measurement systems are required. In a research project, we developed a novel camera-based alternative, which comes with several advantages. The measurement system consists of a mechanical cross slide that guides an industrial camera over a surface to be measured. Images taken by the camera are used for 3D reconstruction. Finally, the reconstructed point clouds are used to estimate roughness. In this article, we present our measurement system (including the hardware and the self-developed software for 3D reconstruction). We further provide experiments to camera calibration and evaluation of our system on concrete specimens. The resulting roughness estimates for the concrete specimens show a strong linear correlation to reference values obtained by the sand patch method.

## 1. Introduction

The roughness of building materials plays an important role in the field of civil engineering. For example, for the renovation of concrete components, a certain roughness is needed to ensure bonding between the coating material and base concrete [1,2,3,4]. Besides specific adhesion—which involves the physical, chemical, and thermodynamic interactions of surfaces—mechanical adhesion is also crucial for a good adhesive bond [5]. Mechanical adhesion supposes that the applied liquid coating material flows into the holes and gaps of the base concrete, hardens, and anchors like dowels or snap fasteners. Moreover, a higher roughness of the base concrete causes a larger composite surface, which leads to a stronger bond as well. Another benefit of determining roughness is the facilitation of estimating the required amount of coating material. The rougher the surface of a concrete component, the more coating material is required to cover the entire surface.

The most widely used and standardized method for the determination of concrete surface roughness is the sand patch method introduced by Kaufmann [6]. Here, the user applies a pre-defined amount of sand with a standardized grain size onto the surface and distributes it evenly in circular movements. The diameter of the resulting sand patch relates to surface roughness. Despite its simplicity, this method has its shortcomings. For instance, the manual execution requires some experience on the part of the user. Studies have shown that the measured roughness can vary by around 20% depending on the user [7]. In addition, this method does not provide reproducible results. Specifically, the repeated execution of the sand patch method on the same area results in varying diameters of sand patches thus leading to a range of different estimates for roughness. Furthermore, it requires direct contact with the surface to be measured. However, direct contact can lead to wear and modify the original structure of the surface. This in turn can lead to falsified results. Another crucial limitation is the lack of applicability on highly slanted surfaces or ceilings. For these reasons, it would be helpful for many applications to develop more sophisticated and contactless measurement systems for the topographic analysis of technical surfaces.

For application in the construction industry, the system must be deployable on construction sites. Thus, the measurement device has to fulfill a range of requirements. In practical use, building elements usually have to be measured that are already installed and in service. Thus, it is not feasible to detach the elements in order to measure the surface roughness at a stationary system (e.g., in a lab). Consequently, the mobility of the measurement device is crucial. Furthermore, the measurement of strongly slanted surfaces or ceilings requires the device to be hold on the surface by the operator during the measurement for a certain period. Hence, the device also has to be as lightweight as possible. In addition, the simplicity of the commonly used sand patch method has made it very popular and widely used in the construction industry. Accordingly, for progressive measurement methods it is a huge challenge to become established in practice. In order to achieve acceptance, the system must be usable without major technical instructions such as the sand patch method. Finally, to make the measurement device as attractive as possible for users, the costs need to be kept low.

Although there are already many systems and recent developments regarding area-based surface reconstruction (see Section 2), they often do not satisfy the aforementioned requirements. Most of these systems are based on optical methods (such as white-light interferometry, confocal 3D laser scanning microscopy, or focus variation microscopy) and rely on a complex and costly measurement setup or equipment. Further, they are not portable and hence not applicable on construction sites. Our objective is to tackle this gap.

In this paper, we introduce a novel camera-based measurement system for the roughness determination of technical materials, such as concrete elements. The hardware of the system primarily consists of a cross slide with a controlling unit and propulsion, which guides a monocular industrial camera over the surface to be measured. Images taken by the camera are used for digital 3D reconstruction. The self-developed software for 3D reconstruction mainly involves a two-step image matching algorithm: Structure from Motion (SfM) and Dense Image Matching (DIM). Finally, the reconstructed dense point clouds are used for the estimation of roughness.

Our measurement system is designed to satisfy the aforementioned requirements and provides additional advantages compared to other methods. The novelty of our approach can be related to the combination of the following features, which is unique in this form:Fully digital measurement system and reproducibility of results.Contactless and area-based measurement.Deployable on construction sites and high mobility.Applicability on arbitrary oriented surfaces.Easy to use.Lightweight.Low-cost.

Following this introduction, this paper is organized as follows: Section 2 gives an overview of recent developments regarding methods for measurements on concrete surfaces. In Section 3, we provide some necessary fundamentals, in particular a brief definition of roughness and the basics of digital photogrammetry. Subsequently, in Section 4, the developed measurement system is introduced. This includes the concept for capturing the images of object surfaces, the hardware used to build the prototype, and the custom-built 3D calibration test-field. In Section 5, the methodology for 3D reconstruction and roughness estimation is provided. Following this, Section 6 covers our investigations into camera calibration and trials at estimating the roughness of 18 concrete specimens. Section 7 presents and discusses our results. Finally, in Section 8, we present our conclusions with an additional insight into future work. 

## 2. State of the Art and Related Work

A simple modification of the sand patch method was introduced in [8], which enables the measurement procedure to be performed on arbitrarily oriented surfaces. Instead of pure sand, they propose a paste consisting of two parts of sand and one-part medical ultrasound gel. The measurement procedure is carried out in a similar way to the traditional sand patch method. Even though it is applicable to arbitrarily oriented surfaces due to the stickiness of the paste, it still suffers from the aforementioned disadvantages. Furthermore, after the measurement, additional effort must be expended to remove the paste from the surface.

Besides the frequently used sand patch method, there is also the Digital Surface Roughness Meter (DSRM) [4,9] widely used in practice for roughness determination. These devices are either laser-based or based on mechanical stylus and capture the surface in both cases in a line-based manner. Although they are usually small, handy, and low-cost, they also hold some disadvantages. The stylus-based ones measure by a stylus tip tracing the surface, which can lead to wear of the stylus or the object surface and furthermore is limited by the radius of stylus tip. Even though the laser-based ones do not suffer from this disadvantage, they are still limited to line-based assessment. 

The ASTM E 2157 Circular Track Meter [10] represents a more sophisticated device compared to DSRM and covers the surface in a circular way. It is laser-based as well and thus also provides non-contact measurements. However, it also suffers from some shortcomings. The surface is covered only by a circular profile and hence it provides (like DSRM) only profile-based assessments. Moreover, even though it is portable and relatively small with a size of 40 cm × 40 cm × 27 cm, it weighs about 13 kg, and thus does not provide the applicability on highly slanted surfaces or ceilings (unless it is held by an extraordinarily strong athlete). In addition, the system is like most laser-based systems comparatively expensive.

In terms of building survey, Terrestrial Laser Scanners (TLS) and Mobile Laser Scanners (MLS) are gaining importance in the construction industry. However, although TLS and MLS are area-based and contactless measurement systems they typically provide geometric accuracies in the range of a few millimeters only [11,12,13] and are therefore not suitable for roughness determination. 

Recently, laser-based triangulation methods have been added to international ISO standards as an alternative to the sand patch method [14]. The basic principle of laser triangulation methods is that either a laser point or laser line is projected by a laser diode onto the surface of an object. That point or line is detected by a position sensitive detector (PSD), which is placed at a fixed distance and angle to the laser diode. In this way, a change in the distance between the laser diode and object surface results in a change of the signal position on the PSD. Finally, the depth of the object can be determined by trigonometry.

Three laser-based triangulation systems for determining the roughness of concrete surfaces were introduced by Schulz [15,16,17,18]. Two of the custom-built systems, Profilometer and ELAtextur, use a point-based laser sensor, which is either mounted on a linear actuator and moved linearly over the surface (Profilometer) or rotated on a vertical axis (ELAtextur). The third measurement system is a laser sensor projecting a line instead of a single point. The length of the line is 100 mm and consists of 1280 single points. The results of the laser-based triangulation systems are strongly correlated with results obtained using the sand patch method.

Werner et al. [19] compared two off-the-shelf laser-based triangulation systems in terms of determining the surface parameters of concrete fractures. One is the cost-efficient system DAVID 3D and the other one is the high-end system LEICA T-Scan. They divide the surface of fractured concrete into different scales—micro, meso, and macro level. In the study, it is found that the DAVID 3D system is suitable for measuring at the meso level, which is specified as consisting of features measuring between 1 and 100 mm across. However, for smaller features they recommend the LEICA system. For the scale-independent determination of surface parameters they consider the fractal dimension. 

Laser-based triangulation techniques are more sophisticated and permit contactless, user-independent, and reproducible measurements, unlike the sand patch method, but they provide, in general, only a point-based or line-based measurement. In addition, professional laser-based systems are comparatively expensive.

Image-based measurement methods such as digital photogrammetry, on the other hand, represent a competitive alternative to laser-based methods since only a camera is required. While close-range photogrammetry is generally used for 3D measurements of objects ranging in size from several millimeter to several meter, this method is suitable for measurements in micrometer range as well. In traditional digital photogrammetry, before the measurement process, objects to be measured usually have to be prepared by targets, which are later used to measure these in the images. This is often done manually or semi-automated, as, for example, in the software PHIDIAS [20].

For instance, digital photogrammetry was used to monitor cracks in concrete elements [21,22]. In the experiments, the surfaces of the concrete elements were covered by a grid of targets. When continuously increasing stress is applied to the concrete elements, the resulting cracks move the targets. Through these movements, the origin and the evolution of the cracks were observed. However, target-based measurements necessitate some preparation effort. In addition, just a sparse set of points can be measured.

However, in recent decades, the processing power of computers has increased immensely, allowing the development of powerful algorithms for fully automatic feature point detection. By using these feature detectors [23,24,25,26], there is no need for targets to be installed. Targets can be used in this case for georeferencing purposes or if measurements with particularly high accuracy are required. Moreover, if a monocular camera is used, units used will be dimensionless. In this case, targets can also be used to determine scale. Furthermore, algorithms for dense image matching allow measurements for every pixel thus achieving dense surface measurements.

A study that compares laser triangulation, photogrammetry, and the sand patch method with each other has been published by Wienecke et al. [27]. These three methods result in three different roughness coefficients. Although a comparison of the measured values indicates some correlation in the results of these three methods, there are still deviations evident. As possible reasons for these deviations, the authors mention, among other reasons, the lack of reproducibility of the sand patch method and the difficulty associated with measuring certain types of surface due to relatively large grain size used in the sand patch method.

In the literature, there are many other examples concerning optical profilometry for the characterization of surface texture. These methods are often based on white-light interferometry, confocal laser scanning microscopy, or focus variation microscopy. 

For instance, fringe-based laser interferometry was used in [28] to investigate the concrete substrate roughness in patch repairs. To be precise, the authors examined two different methods for removing defective concrete, i.e., electric chipping hammers and Remote Robotic Hydro-erosion (RRH). A total of 60 slab specimens were analyzed and, for each of them, four different roughness parameters were calculated based on profile lines. The results reveal that RRHs are able to create rougher surfaces than chipping hammers. The main advantage of the proposed system is that it captures the surface in an area-based way and creates a detailed and accurate 3D topography of the surface. 

Confocal microscopy was applied in [29] to characterize the fracture surface of six specimens of Portland cement pastes and mortar. The investigated specimens contain two types of sand, fine sand with an average diameter of 0.15 mm and coarse-grained sand with an average diameter of 0.75 mm. They estimated the surfaces of the specimens with a depth resolution of 10 µm and a total range for the depth of about 200 µm for the paste and 800 µm for the mortar specimens. The roughness value is finally determined by the ratio of the actual surface area and the nominal surface area. The results reveal that the actual surface area of cement pastes are about 1.8 times greater than the nominal surface area, while for mortars it is between 2.4–2.8 times the nominal surface. Further, it is found that the roughness is not simply a function of the largest aggregate size. 

A range of other non-destructive methods (e.g., impulse response (IR) and impact-echo (IE)) are presented and applied for testing surface morphology of concrete substrates [30]. The equipment used by both IR and IE methods are portable due to their low weights of around 1 kg. However, the main drawbacks are the contact- and line-based assessment of the surface. Furthermore, the results are sensitive to mechanical noise created by equipment impacting. 

In our prior research, we have conducted feasibility studies for the determination of the surface roughness of concrete using photogrammetry [31]. In the measurement setup, a digital single-lens reflex camera (DSLR) with a macro-objective lens was used in order to capture the surface of a particular concrete specimen. The reconstructed point cloud is visualized as a height map, showing the peaks and valleys of the concrete surface.

Building on our previous research, we propose that photogrammetry can also be used for analyzing the surface of building materials such as concrete elements, since it allows for a large-scale investigation of object surfaces with high precision. In particular, image-based methods provide an area-based measurement of the surface in contrast to some of the aforementioned methods. Additional benefits are the contactless and therefore non-destructive testing and the repeatability of results. Furthermore, a camera-based approach has the advantage of being portable and can be used on arbitrarily oriented surfaces. 

## 3. Theoretical Background

In the following sections of this chapter, we provide some fundamentals that contribute to the understanding of the succeeding content of the paper. First, a brief definition of roughness is introduced by classifying it into orders of shape deviations and providing common roughness parameters. Subsequently, basics of digital photogrammetry are covered. 

### 3.1. Defining Roughness

#### 3.1.1. Shape Deviations

According to the German standard DIN 4760 [32], the deviation between the actual surface of an object, which is captured by a measurement instrument, and the geometric ideal surface is defined as shape deviation. Shape deviations are further classified into a total of six orders, as shown in Table 1.

While the first two order types specify the form deviation and waviness of surfaces, types 3–5 describe the properties of a surface that are relevant for our purposes, specifically the roughness. However, the actual surface of a technical material consists of all orders of shape deviations. Figure 1 shows an example of a surface profile composed of multiple types of shape deviation orders.

#### 3.1.2. Parameters

A range of parameters are available for the description of the roughness of technical surfaces and are specified in the standard DIN EN ISO 4287 [33]. Typically, most of these parameters are designed for single profile lines. In the following, a small selection of the most widely used parameters will be introduced and briefly discussed. 

##### Arithmetical Mean Deviation of the Assessed Profile ($R_{a}$)

The arithmetical mean deviation $R_{a}$ of an assessed profile is calculated by integration of the absolute values of the profile deviations along a reference line and dividing the sum by the length of the line. Thus, the parameter is calculated as follows:(1)$$R_{a}=\frac{1}{l}\overset{l}{\underset{0}{\int }}\left|Z\left(x\right)\right|dx.$$

Hence, $R_{a}$ corresponds to the average distance of the profile line with respect to the mean line, as illustrated in Figure 1 using an exemplary profile line.

However, there are some drawbacks associated with using $R_{a}$ to describe surface roughness with regard to adhesive properties. To demonstrate the problem, two profile lines with plainly different surface textures are shown in Figure 2. For simplicity, both lines are sketched as rectangular functions with the same height of rectangles. While the left profile line consists of less rectangles with a larger width, the right one is composed of more narrow rectangles. Due to the higher area of contact, basically the length of the profile line, the surface represented by the right line would lead to a greater adhesive bond. However, the arithmetical mean deviation is the same for both these cases.

Accordingly, arithmetical mean deviation is an inappropriate parameter for determining adhesive behavior but is suitable for the estimation of the necessary amount of coating material to completely cover the surface.

##### Mean Texture Depth (MTD)

Principally, the occurring sand patch after performing the sand patch method can be regarded as an approximation for a cylinder with an irregular depth (or height respectively). With the knowledge of the volume V of the applied amount of sand and the diameter d of the sand patch, the mean height h—in terms of the sand patch method also referred as mean texture depth (MTD)—can be calculated using the rearranged formula for the volume of a cylinder. The mean texture depth can thus be determined in the following way:(2)$$MTD=\frac{4\cdot V}{\pi \cdot d^{2}}.$$

To simplify, MTD corresponds to the average distance of the object surface to a plane deliminated by the highest peaks. An illustration of mean texture depth is shown in Figure 3.

### 3.2. Digital Photogrammetry

#### 3.2.1. Camera Model

The principle of photogrammetric imaging is based on a simplified model: The pinhole camera. When an object point is recorded by a camera, the optical ray from the object point runs straight through the optical center of the camera and is projected onto the image sensor as an image point. Thus, the optical center of a camera is the mathematical point in space through which all the optical rays of the captured object points pass. Accordingly, this imaging procedure is also known as central projection and can be described mathematically with the collinearity equations (see [34,35]).

The pinhole camera model, however, is just an idealized mathematical model of photogrammetric imaging. The physical model also plays an important role. The photographic objective lens of a camera usually consists of several different single lens elements through which the incoming optical rays are refracted several times before they come up onto the image sensor. The refraction through the lenses or an asymmetrical structure inside the photographic objective causes the straight lines of a viewed object to appear curved when projected onto the image sensor. These aberrations, also called lens distortion, consist mainly of radial and tangential components, and can be modeled, for example, with polynomial correction functions. While the influence of radial distortion increases depending on the distance of a pixel to the principal point, tangential distortion increases asymmetrically. To model photogrammetric imaging while considering lens distortion, additional modeling terms are applied to the collinearity equations. At this point, we do not go into further detail and refer to the relevant literature (see, inter alia, [34,35]).

The interior orientation of a camera describes the relationship between the image plane and the optical center, specified by the position of the principal point, the focal length, and the lens distortion coefficients. In contrast, exterior orientation refers to the pose (i.e., position and orientation) of a camera with respect to a world reference frame. 

#### 3.2.2. Epipolar Geometry

Given just one image with known pose and interior orientation of the camera, only the direction of a 3D world point corresponding to a certain image point can be determined. Schematically, for each image point, a ray can be generated that starts from the optical center of the camera and goes through the respective point in the positive image plane. The corresponding world point is then located somewhere on the ray.

However, to determine the actual spatial position of a world point, a second image, in which the same world point is captured from a different point of view, is required. Figuratively speaking, the optical ray of the first image is then depicted as a line—also called epipolar line—in the second image. Thus, given an image point in the first image, the location of its correspondence in the second image is restricted on its epipolar line. After identifying the corresponding image point, another optical ray for the same world point can be generated from the second image. The sought world point is then ideally located at the intersection of both rays.

The geometry between two images of the same scene is called epipolar geometry and can be encoded by the Fundamental Matrix F. In general, F maps an image point of the first image to its epipolar line in the second image. While F works with pixel coordinates, the Essential Matrix E, which is a specialization of F, deals with calibrated cameras and uses normalized image coordinates.

In general, the images are either divergent or convergent, which results in the epipolar lines running oblique. However, after rectifying both images to the same virtual plane and thus transforming to the canonical stereo configuration, the epipolar lines become parallel and corresponding image points lie in the same image row (or in the same column in the case of vertical stereo images). Figure 4 shows an image pair before and after a stereo rectification.

The shift—also known as the disparity—of two corresponding pixels in a rectified stereo pair can be encoded in a disparity map. The shorter the distance of a scene point to the camera, the greater the disparity of the pixel pair. For very distant objects, there is basically no displacement and thus the disparity approaches 0. Consequently, disparity is inversely proportional to depth. 

## 4. Measurement System

### 4.1. Concept for Image Capture

Our 3D reconstruction procedure is based on image matching and requires photogrammetric images of the surface to be taken. Essentially, the interior orientation of the camera has to remain constant during the whole recording time and the scene should be captured from different points of view with a sufficiently high overlap. 

As recording geometry for capturing images of object surfaces, we adapted the concept of traditional aerial photogrammetry since it allows a simpler hardware construction. For a more accurate measurement of the surface (especially in depth), the camera would have to capture the surface from different directions (not only with viewing direction perpendicular to the surface). However, to accomplish this, the hardware of the system would have to be designed much more complex (e.g., in order to tilt the camera to different directions during the capture). Since our aim is to develop a simpler, user-friendly, maintenance-low, lightweight, and cost-efficient measurement system, such a complex hardware construction would be a disadvantage in our case. 

Since each object point has to be captured by at least 2 images to determine the 3D coordinate, at least 50% of overlap is required in the images to reconstruct the surface without gaps. A higher overlap, though, leads to over-determination (since in this case the surface will be captured by more than 2 images). This in turn leads to better estimates of the 3D object point coordinates. We use an overlap of 60–80%, which is common in aerial photogrammetry. 

Accordingly, the measurement camera is moved in a meandering path parallel to the object surface. Further, the orientation of the camera remains constant with perpendicular viewing direction to the surface. Strongly overlapping images are taken in evenly spaced intervals. An illustration of our concept of the recording geometry using a monocular camera is depicted in Figure 5.

### 4.2. Apparatus

The prototype of the measurement system consists of a cross slide with a controlling unit and propulsion, which guides an industrial camera in a meandering path automatically over the surface to be measured. The camera is moved on two axes and images are taken in stop and go mode. Subsequently, the captured images are transferred to a measurement computer via USB. Recording time depends on the number of images and the configured degree of overlap. It currently takes 30 images in around 5 min (5 images in x-direction and 6 images in y-direction). We use a Basler acA3800-14um monocular industrial camera (Ahrensburg, Schleswig-Holstein, Germany). Some main camera specifications are shown in Table 2.

Provided that the surface is not too inhomogeneous, evaluating an area of 10 cm × 10 cm should be sufficient to derive representative results for the whole surface of a concrete element. Since for an area to be reconstructed and subsequently evaluated, it has to have been captured by at least 2 images and with a sufficiently large baseline, an area of almost 20 cm × 20 cm has to be captured.

The measurement system can be powered either by an external power supply or by a built-in rechargeable battery. In addition, to ensure uniform, diffuse illumination of the area to be captured, LED strips are attached all around the perimeter of the apparatus. The intensity of the light can be dimmed smoothly. Consequently, the quality of measurement is independent of ambient illumination. Figure 6 shows the measurement system prototype. 

### 4.3. Custom-Built 3D Calibration Test-Field

For a sufficiently accurate calibration of the industrial camera, we designed a 3D calibration test-field with proper spatial distribution of the ground control points (GCPs). The custom-built calibration field consists of a 17 cm × 17 cm plate with a total of 144 columns on top. The columns have 3 different heights and are arranged with alternating heights in order to guarantee a proper spatial distribution of the GCPs. The appropriate spatial distribution ensures that in each of the images, GCPs from different depths are captured. The GCPs are distributed on the calibration field so that on each column there is one uncoded target and on the ground of the plate there are a total of 121 coded targets. In this way, the maximum height difference of the points is 2.25 cm. Due to the comparatively small size and the required fineness, the calibration field was manufactured by 3D printing. The required model for this was constructed using CAD software.

Additionally, four calibration rods with targets on each end of the rods are attached around the calibration field. These are used to determine the scale. The distances between the pairs of targets were measured by an interference comparator. Calculating the mean of the standard deviations of all 4 pairs leads to around 7 µm. The printed 3D calibration test-field, including the calibration rods, is shown in Figure 7.

## 5. Methodology

### 5.1. 3D Reconstruction Pipeline

Although there are already software applications, which perform photogrammetric pipelines for 3D reconstruction (such as Agisoft Metashape [36]), we developed our own workflow, which fulfills our specific needs and furthermore is independent of commercial products. Hence, to perform 3D reconstruction of object surfaces using images taken by the calibrated industrial camera, we implemented an image matching pipeline, which includes both Structure from Motion (SfM) and Dense Image Matching (DIM) procedures. We use the SfM algorithm to refine the pose of the camera, which is roughly pre-determined due to the configured recording geometry. After determining the exact poses of the camera, we use DIM to obtain disparity maps of all image pairs and further, using these, we generate a point cloud of the scene as dense and gapless as possible. Our complete pipeline for incremental 3D reconstruction is presented in the following diagram (Figure 8). 

#### 5.1.1. Preprocessing

Initially, the pipeline requires as input overlapping images of the object surface and camera calibration parameters, such as the interior orientation of the camera and lens distortion coefficients. Afterwards, all images are preprocessed to remove lens distortion and thus allow for the use of the simplified pinhole camera model. 

#### 5.1.2. Structure from Motion

An illustration of the single steps of the SfM procedure is provided in Figure 9. 

Interest operators, as, for instance, the widespread SIFT [25] and SURF [26], identify distinctive points in the images. These points—also called feature points—are basically corners, edges, or blobs, which lead to significant changes in the intensities of pixels and thus differ strongly from their neighborhood. Once the location of a feature point is detected, an image patch around the point is extracted, transformed in a manner so that it becomes scale- and rotation-invariant and stored in a vector called feature descriptor. In this way, every feature point in an image gets its own fingerprint, which can further be used to distinguish one point from the others. We use SURF as interest operator for feature detection and subsequent extraction since it provides around 3 times higher speed with similar accuracy when compared with SIFT [37]. 

Subsequently, matching of extracted feature descriptors in distinct images is utilized for identifying pairs of images looking at the same part of the scene as well as estimating the relative orientation between the image pairs. The relative orientation of an image pair is defined by the orientation, represented by a rotation matrix R, and the position, represented by a translation vector t, of the second image with respect to the first one. Since the exterior orientation of the camera is approximately pre-determined and basically corresponds to the case of aerial photogrammetry, we can exploit some key conditions. For example, the regularly spaced and strip-wise arranged images allow simplifications for establishing image pairs. Hence, we simply pick the first two images as an initial pair and consider only consecutive images for setting up further pairs. Provided that the system is configured with a sufficiently small movement of the camera between images, a high overlap is guaranteed. Thus, it can be assured that all images share common feature points, which prevents the occurrence of multiple models. Another benefit of the pre-known image sequence is that we neither have to deal with a possibly bad initialization of the model, which in turn could lead to a bad reconstruction due to the accumulation of errors. 

The formation procedure of image pairs further runs through a multi-stage filtering step in order to achieve as robust matches as possible. Given an image $I_{i}$, we compare it with all previous images $I_{i-j}$ and search for the two nearest matches of each feature point in the image $I_{i}$ to that in the image $I_{i-j}$ and vice versa. The first filtering step involves a distance ratio test as proposed in [25]. Accordingly, a match is considered poor if the distance ratio between the first-best and second-best match falls under a particular threshold. Following, we also eliminate matches based on a symmetry criterion and hence reject those that have not matched each other. The last filtering step involves a geometry verification process in which we use the RANSAC test for estimating the fundamental matrix based on the remaining point matches and identify possible outliers. Finally, an image pair is considered as valid with enough overlap if the number of filtered point matches exceeds a pre-defined threshold, e.g., in our case, 100. The diagram for establishing image pairs including the filtering procedure is shown in a box in Figure 10. The box basically represents a zoom-in of the fifth box (Form image pairs) in the diagram of the 3D reconstruction pipeline (Figure 8). 

Based on the filtered point matches, the fundamental matrix F is computed as proposed in [38], and further, with a multiplication by the camera intrinsic matrix K, we obtain the essential matrix E. Subsequently, E is decomposed using singular value decomposition (SVD) into a rotation matrix *R* and a translation vector *t*, which represents the relative orientation between the image pair. Due to the unknown scale between both images, t is normalized. 

Given the relative orientation of an initial image pair, we first triangulate corresponding points in space in order to determine the 3D coordinates of the associated world points. However, due to noise, the rays will never truly intersect. To solve that problem, we apply linear triangulation [39] for minimizing the algebraic error and thus obtain an optimal solution in the least squares sense.

Consecutive images are successively registered in the local model coordinate system utilizing 2D- to 3D-point correspondences as described in [40]. Therefore, with the correspondences of 2D feature points of a considered image and 3D world points triangulated by previous image pairs, we solve the perspective-n-point (PnP) problem [41]. As a result, we obtain the exterior orientation of the considered image with rotation matrix R and translation vector t in the given coordinate system. Finally, the new feature points of the registered image are also triangulated with those of the previous images. 

Obtained results for the exterior orientation, world points, and feature points are considered to be approximations and subsequently optimized in the course of least-squares bundle adjustment. Thus, after each image registration, we apply Levenberg–Marquardt optimization [42,43] in order to simultaneously refine the exterior orientation of the images, the coordinates of the triangulated world points and the feature point coordinates. However, the interior orientation of the camera is considered fixed since it has been calibrated with high accuracy beforehand. 

#### 5.1.3. Dense Image Matching

Contrary to SfM that estimates 3D coordinates only from distinctive image points, the objective of DIM is to find the correspondence preferably for *each* pixel. These correspondences are then further used for triangulation in order to determine the 3D world coordinates.

Generally, algorithms for stereo matching can be divided into local and global matching methods. Local methods, as the name indicates, consider only local windows for the correspondence search. Specifically, for a certain pixel in the reference image, the window is sliding over a range of possible correspondences in the target image, restricted by a pre-defined disparity search range d. Matching costs, based on some similarity measure, are calculated for each possible correspondence. The correspondence is finally chosen as the pixel providing the minimum cost.

Local matching methods are in general comparatively easy to implement and very computational and memory efficient. However, disparity selection using a local Winner-takes-all (WTA) strategy struggles with point ambiguities like poor or repetitive textures and thus often leads to many mismatches.

Global matching methods, for example, based on Belief Propagation [44] or Graph Cuts [45], on the other hand, set up an energy function that takes all image points into account. An optimal disparity map is achieved by minimizing the energy function. Due to the inclusion of all image points in non-local constraints, global algorithms usually achieve a higher accuracy than local methods, but at the cost of a higher computational complexity.

In contrast, Semi-Global Matching (SGM), introduced by Hirschmuller [46], represents a compromise solution since it offers a reasonable trade-off between accuracy and computation complexity. SGM also sets up a global energy function E, including a data term $E_{data}$ and a smoothness term $E_{smooth}$. The former measures the pixel-wise matching cost while the latter is for consistency, accomplished by penalizing discontinuities in the disparity map. The disparity map generation is performed by minimization of E over several one-dimensional paths using dynamic programming.
(3)$$E\left(D\right)=\underset{p}{\sum }\left(\underset{E_{data}}{\underbrace{C\left(p,D_{p}\right)}}+\underset{E_{smooth}}{\underbrace{\underset{q\epsilon N_{p}}{\sum }P_{1}\cdot T\left[\left|D_{p}-D_{q}\right|=1\right]+\underset{q\epsilon N_{p}}{\sum }P_{2}\cdot T\left[\left|D_{p}-D_{q}\right|>1\right]\;}}\right).$$

In order to meet real-time requirements, we implemented the compute-intensive DIM, specifically SGM, for Graphics Processing Units (GPU) using Nvidia’s programming model CUDA [47]. We parallelized each sub-algorithm of SGM for massively parallel processing and implemented parallel executable functions—also called kernel—for each of the algorithms. In the following, we recall the main steps of the SGM algorithm and, building on that, we introduce our parallelization scheme for GPU implementation.


ACost Initialization:


An initial cost volume $C\left(p,D_{p}\right)$ of size w×h×d has to be built up, where w and h are the width and height of the image and d is the disparity search range. For that purpose, pixel-wise matching costs based on some similarity measure have to be calculated for each pixel p in the reference image to its d potential correspondences in the target image.

The most widely used similarity measures are sum of absolute differences (SAD), sum of squared differences (SSD) and normalized cross correlation (NCC) [48]. Even though SAD and SSD are easy to implement and real-time capable, they assume that pixel intensities of the same object are almost equal in the images, which might not be always true. In contrast, NCC is less sensitive to changes of intensities as well as to gaussian noise, however it is computationally intensive.

In contrast, Census Transform (*CT*) [49] followed by a Hamming distance calculation represents an outstanding approach. *CT* compares the intensity of a considered pixel with that of its neighbors to generate a binary string—commonly referred as *CT* feature—as a representation for that pixel. The advantages of *CT* are illumination invariance, efficiency, and ease of implementation due to simple operations as XOR. We use a slightly modified version of *CT*—the Center-Symmetric Census Transform (*CS*-*CT*)—as proposed in [50], which compares neighboring pixels only to each other that are mirrored at the considered pixel. Therefore, the total number of operations decrease by around 50% and a more compact representation using just half of the memory is provided. The calculation of a *CS*-*CT* feature for a pixel at the location (x,y) is given as:(4)$$\begin{array}{c} CS-CT_{m,n}\left(x,y\right)=\underset{\left(i,j\right)\in L}{⊗}s\left(I\left(x-i,y-j\right),I\left(x+i,y+j\right)\right),\;\\ \mathrm{with}\;s\left(u,v\right)=\{\begin{array}{c} \begin{array}{cc} 0,\; & if\;u\leq v \end{array}\;\\ \begin{array}{cc} 1, & otherwise \end{array} \end{array}. \end{array}$$

Subsequently, the matching cost of two pixels is given by the Hamming distance of their *CT* features and is represented by:(5)$$C\left(p,q\right)=Hamming\left(p,q\right)=\|CS-CT_{L}\left(p\right)⊕CS-CT_{R}{\left(q\right)\|}_{1}.$$

We implemented a kernel for *CS*-*CT*, which operates per image. The kernel is executed by a two-dimensional layout of threads in order to assign one pixel to each thread in the simplest possible way. The window size is chosen with 7 × 9 pixel and has the advantage that the resulting bit string of a *CS*-*CT* feature has a size of 31 bit and thus fits into a single 32-bit integer data type. Accordingly, an image with a size of 3840 × 2748 pixel leads to a total memory consumption of 40 MB.

A second kernel works on two *CS*-*CT* transformed images and calculates the matching costs. The kernel is again based on a two-dimensional thread layout with each thread calculating the Hamming distance for a specific *CS*-*CT* feature of the reference image to all d potential candidates in the target image. The XOR operation for two features and the subsequent population count in the resulting bit string—basically the number of ones—lead to values ranging from 0 to 31. Therefore, the use of a 1-byte data type for the Hamming distance and a disparity range of 80 values results in a total memory consumption of 805 MB for the initial cost volume $C\left(p,D_{p}\right)$.


BCost Aggregation:


The costs of the initial cost volume have to be aggregated over several paths considering two penalties P1 and P2, with P2>P1. The recursive aggregation on a path r at a specific pixel p and disparity value d is defined as:(6)$$\begin{array}{c} L_{r}\left(p,d\right)=C\left(p,d\right)\\ +\;min\left(L_{r}\left(p-r,d\right),L_{r}\left(p-r,d-1\right)+P1,L_{r}\left(p-r,d+1\right)+P1,\underset{i}{\min}L_{r}\left(p-r,i\right)+P2\right)\\ -\;\underset{k}{\min}L_{r}\left(p-r,k\right). \end{array}$$

Summing the aggregated costs over all paths delivers the aggregated cost volume S with:(7)$$S\left(p,d\right)=\underset{r}{\sum }L_{r}\left(p,d\right).$$

For the aggregation of the cost volume C, we implemented only the horizontal and vertical paths $L_{r}$ in order to keep the computing time low. Thus, the aggregation is performed in the 4 directions left-to-right, right-to-left, top-to-bottom, and bottom-to-top. However, the recursive calculation prevents parallelization in each path. Though, we exploit the independency of different paths in the same direction (in the case of the horizontal paths) and different columns (in the case of vertical paths). 

For instance, a kernel was implemented for the aggregation in the direction left-to-right and can be called with a specific column number. In that column, the parallelization occurs in the rows and disparity space of the cost volume. In this way, each thread is responsible for the aggregation of the cost in a particular row and disparity. The kernel is called for every column starting from the left to the right, with the first column being initialized by the initial matching cost. The procedure is analogous for the three other kernels of the remaining path directions. 

Subtracting the minimum path cost $\underset{k}{min}\;L_{r}\left(p-r,k\right)$ in each step of the cost aggregation limits the maximum cost in a path with $C_{max}+P2$. Thus, the aggregated costs never exceed the range of 16 bits. Using images with a size of 3840 × 2748 pixel, a disparity range of 80 values and a data type of 2 bytes for the aggregated costs, the total memory consumption can be estimated at around 1.6 GB for each path.


CDisparity Selection:


For a simple case, the disparity d for each pixel p can finally be chosen based on the minimum cost in the aggregated cost volume. However, the subsequently generated point cloud based on that disparity map shows banding artifacts, since there are only 80 possible disparity values, which in turn leads to only 80 different coordinates for the depth. 

Hence, we use sub-pixel interpolation, as proposed in [51], for choosing the disparity values in order to obtain a steady and continuous point cloud. Accordingly, the disparity with the minimum aggregated cost and both of its neighbors are considered for the estimation of a parabola. The disparity is finally chosen as the abscissa on which the parabola provides its minimum. Thus, the interpolated disparity for a pixel p corresponds to the vertex of the parabola and can be expressed by: (8)$$d_{min\_sub}=d_{min}-\frac{\left(S\left(p,d_{min+1}\right)-S\left(p,d_{min-1}\right)\right)}{\left(2\cdot S\left(p,d_{min-1}\right)+2\cdot S\left(p,d_{min+1}\right)-4\cdot S\left(p,d_{min}\right)\right)},$$
where $d_{min}$ represents the integer disparity with the minimum aggregated cost. 

##### Disparity Map Fusion and Point Cloud Generation

The actual depth z of a point can be derived in the following way:(9)$$z=\frac{b\cdot f}{d},$$
with d the disparity of the point, f the focal length of the camera, and b the baseline of the camera between two images. 

However, the obtained disparity maps were generated individually and usually have overlapping areas. Thus, in the case that the disparity maps are triangulated independently, multiple 3D points would be generated for some of the same physical object point. To avoid this, a three-dimensional grid of voxels could be built up for filtering multiple points. However, since our case corresponds to a 2.5D digital elevation model, we build up a two-dimensional regular grid with evenly sized cells parallel to the image planes. The points of the disparity maps are triangulated into these cells. In the case of multiple points falling into a particular cell, we simply pick the median of the depth of the points. Accordingly, for each cell we obtain at most one value for the depth.

### 5.2. Adapting Roughness Parameter to 3D Point Clouds

Reconstructed dense point clouds allow for the analysis of the surface topography. This can be used, for example, to estimate the roughness. Estimation of roughness can be performed (e.g., as in the conventional case) using single extracted profile lines as well as using the entire point cloud for an area-based determination. For this purpose, we adapted existing parameters (see Section 3.1.2) for 3D point clouds.

#### Arithmetical Mean Deviation $R_{a}$

For the calculation of the parameter $R_{a}$, first, the reference plane to which the mean deviation of the points is subsequently calculated has to be determined. In general, the plane needs to be estimated by minimizing the squared Euclidean distance of the measured surface points to the plane. That basically corresponds to the orthogonal distance regression (ODR), which also accounts for errors in the independent variables x and y in addition to the dependent variable z unlike the ordinary least squares (OLS) regression model. However, the main axes of the point cloud are basically parallel to the XY-plane of the coordinate system since the surface is captured parallel to the image plane and hence the difference between OLS and ODR becomes negligible. Consequently, we use OLS to estimate a plane so that the sum of squared vertical distances of the points to the plane becomes minimal.

With the points $p_{i}=\left(x_{i},y_{i},z_{i}\right)$ of the point cloud, we first set up an over-determined equation system for the plane π:(10)$$\underset{A}{\underbrace{\left[\begin{array}{ccc} x_{1} & y_{1} & 1\\ x_{2} & y_{2} & 1\\ & \ldots & \\ x_{n} & y_{n} & 1 \end{array}\right]}}\underset{x}{\underbrace{\left[\begin{array}{c} a\\ b\\ c \end{array}\right]}}=\underset{b}{\underbrace{\left[\begin{array}{c} z_{1}\\ z_{2}\\ \ldots \\ z_{n} \end{array}\right]}}.$$
where a and b are the slopes of the plane in the x- and y-direction and c is the intersection of the plane with the z axis. Subsequently, a closed-form solution of the equation system is determined by multiplying both sides of the equation by the left pseudo inverse of the design matrix A. Thus, we estimate the parameters of the regression plane $\hat{\pi }$ with
(11)$$\left[\begin{array}{c} \hat{a}\\ \hat{b}\\ \hat{c} \end{array}\right]={\left(A^{T}A\right)}^{-1}A^{T}b.$$

Finally, the calculation of the parameter $R_{a}$ is done by numerical integration of the points. Accordingly, we sum the vertical distances of the points $p_{i}$ to the regression plane $\hat{\pi }$, which are basically the residuals ${\hat{e}}_{i}$, and divide it by the total number of points n:(12)$$R_{a}{}_{pointcloud}=\frac{1}{n}\overset{n}{\underset{i=1}{\sum }}d_{vertical}\left(p_{i},\hat{\pi }\right)=\;\frac{1}{n}\overset{n}{\underset{i=1}{\sum }}{\hat{e}}_{i}.$$

## 6. Experiments

In the following, we describe our conducted experiments. Initially, the industrial camera being used was calibrated. Subsequently, the measurement system was assessed using a single concrete specimen. For evaluation of roughness estimation, we further conducted experiments using 18 concrete specimens. An overview of the performed experiments is shown in Figure 11.

### 6.1. Camera Calibration

In order to reconstruct the object surface as accurately as possible, the interior orientation of the camera has to be determined. This consists mainly of the focal length, the position of the principal point in the x- and y-directions, the radial distortion parameters k1–k3, and the tangential distortion parameters p1, p2. In the following subsections, the conducted experiments regarding camera calibration are presented. This involves two different approaches: A preliminary self-calibration and a more elaborate calibration using a custom-built 3D calibration test-field.

#### 6.1.1. Self-Calibration

Performing self-calibration, in which the camera is calibrated without ground control points (GCP), but only with measured image point coordinates of a (flat) concrete surface, leads to unsatisfactory results. The reason for this lies primarily in the lack of spatial distribution of the points and causes the parameters of the interior orientation of the camera to correlate with each other. Table 3 shows the dependencies of the parameters in the form of the correlation coefficients after the self-calibration procedure.

The table shows that the radial distortion parameters (k1–k3) especially correlate with focal length (f). The subsequently reconstructed point cloud using these parameters shows a conspicuous curvature (Figure 12). To visualize the curvature, a single profile line was extracted from the reconstructed point cloud. This profile line, 10-fold scaled in the height direction, is shown at the bottom of Figure 12. The histogram of the height distribution of the point cloud also indicates a smeared normal distribution (Figure 12, right).

#### 6.1.2. Calibration of the Test-Field

In order to determine the GCP coordinates of the custom-built 3D calibration test-field, we attached it to a tripod with a rotatable plate and took images using a digital single-lens reflex camera (DSLR). The DSLR was mounted on a ball joint, which can be moved on two axes parallel to the calibration field. The measurement setup is shown in Figure 13. A total of 69 images of the calibration field were taken systematically from different directions and subsequently processed using the photogrammetric software PHIDIAS (Version 2.21, Langerwehe, North Rhine-Westphalia, Germany) [20]. The targets were measured in the images and afterwards, the 3D world coordinates were determined using bundle adjustment. Bundle adjustment also provided information about image measurement accuracy, which is 0.57 µm or 0.12 pixels in the x-direction and 0.59 µm or 0.12 pixels in the y-direction.

#### 6.1.3. Calibration of the Industrial Camera

For the calibration of the interior orientation of the camera, we took a total of 30 images evenly distributed over the object. Subsequently, bundle adjustment with image measurements of the targets was performed in PHIDIAS, with only the parameters of the interior orientation included as unknowns. A cross validation of the adjusted point coordinates and subsequent calculation of the root mean square (RMS) of the point deviations leads to an object measurement accuracy of 3.40 µm for the x-coordinate, 4.19 µm for the y-coordinate, and 18.42 µm in depth. The conspicuously larger deviation for the z-component is related to the recording configuration and the associated poor intersection geometry of the optical rays. For the parameters of the interior orientation, we obtained the values and standard deviations given in Table 4. All parameters successfully passed a significance test with an error probability of 5%.

Parameter estimation using least squares adjustment based on the Gauss-Markov-Model also provided the correlations of the parameters in the form of the correlation coefficients. These are listed in Table 5. The previously occurring correlations of focal length and radial distortion parameters have been significantly reduced. The remaining correlations between the k-parameters or between p1 and cx as well as p2 and cy are justified in the mathematical-functional models and can therefore not be completely avoided (see e.g., [34]).

The visual impression of the point cloud reconstructed with the new set of parameters for the interior orientation confirms the significantly improved result as well. A curvature of the point cloud can no longer be noticed after reconstruction. For illustration, the point cloud and a profile line extracted from it and 10-fold scaled in height are shown in Figure 14. The histogram of the height distribution of the point cloud now approximates a normal distribution (Figure 14, right).

### 6.2. System Assessment

#### 6.2.1. Test Objects

We used a set of 18 concrete specimens with plainly different surface textures as test objects. The specimens have a size of 40 cm × 40 cm × 10 cm. For each of them, there are reference values for the roughness, determined by the sand patch method, the laser triangulation method, and the paste method [8]. A selection of three specimens with different roughness is shown in Figure 15.

#### 6.2.2. Measurement Procedure

After calibration of the industrial camera using the custom-built 3D calibration test-field, we examined all 18 specimens with our proposed measurement system. In the following, the measurement procedure is described based on a single specimen.

The measurement system is centrally placed onto the concrete specimen and initially a general functionality testing of the measurement system is performed. After switching on the system, the lighting is adjusted so that the surface to be captured is well illuminated. Following, the procedure for capturing images of the concrete surface is started by a controlling software. The images are captured with the following configuration: Five images in the x-direction and six images in the y-direction resulting in the images having an overlap of about 78% in x-direction and 69% in y-direction. The image capture of the surface with that particular configuration is finished in about 5 min. During capture, the images are transferred continuously to the mobile measurement computer (laptop).

After finishing the capture, the evaluation software is started and reads in the image dataset as well as the camera calibration data. In the first main step, the camera pose is refined using SfM. As a by-product, a sparse point cloud of the object surface is provided as well. The total time for this step is about 1 min. Figure 16 (top, left) shows the estimated camera poses with the sparse point cloud. Subsequently, a dense point cloud of the object surface is generated. Since this is the most compute-intensive step of the reconstruction pipeline, it takes about 6 min to complete. The generated dense point cloud and a zoom-in are presented at the top right corner and at the bottom in Figure 16. Finally, the reconstructed dense point cloud of the concrete surface is used to derive a roughness parameter. For this exemplary specimen, we obtain the value $R_{a}=1.065\;\mathrm{mm}$. 

## 7. Results and Discussion

### 7.1. GPU Acceleration of SGM

The GPU-accelerated implementation of SGM was evaluated regarding the runtime. In order to determine the speed-up of the GPU acceleration, we also developed in C++ programming language a pure CPU-based implementation. In both cases, the same algorithms were used.

As a processing platform, we used Nvidia’s graphics card GeForce GTX 1080 Ti. This high-end GPU contains 28 Streaming Multiprocessors (SM) with 128 CUDA Cores per SM and therefore provides a total of 3584 CUDA cores for parallel processing. The graphic card’s throughput is around 11.70 TFLOPS and it has a memory space of around 11 GB.

As test data, we used stereo images of a concrete specimen previously captured by the measurement system. The area for dense image matching of the stereo images is restricted to the part of the images that is visible in both. Hence, we investigated our pipeline of SGM with grayscale images with a size of 2887 × 2652 pixels. The disparity search range was fixed to 80 pixels.

The comparison of the total runtime between the pure CPU implementation and the GPU-accelerated implementation is shown in Figure 17. We ran both implementations 10 times and charted the average. The runtime of the CPU implementation is therefore around 209 s, whereas the GPU-implementation achieves 4.4 s for the execution. Hence, with GPU acceleration, SGM is completed in only 2.1% of the time needed by the pure CPU implementation.

For a more detailed comparison, we further plotted the runtime of every main algorithm of the SGM pipeline in Figure 18. The chart shows that basically each algorithm benefits from GPU acceleration. This is because the algorithms used by SGM are suitable for parallelization.

### 7.2. Comparison of the Results of Our Measurement System with the Sand Patch Method

The 18 concrete specimens were used to compare the results of our proposed measurement system with those obtained using sand patch method. For this purpose, we plotted our estimated results for the parameter $R_{a}$ and the reference values for the MTD parameter determined by the sand patch method in Figure 19.

The values for the parameter $R_{a}$ estimated by our measurement system are lower than the reference values for the MTD parameter. However, that was to be expected: Although both parameters represent the mean distance of the actual surface to a reference plane, the two reference planes used by the two methods are different. To be specific, $R_{a}$ refers to the mean plane estimated through all points of the surface, while MTD refers to the plane placed onto the uppermost peaks (which leads in general to higher distances between the surface and the reference plane).

Furthermore, a closer look at the measurement values exposes some correlation between both methods. To show this more clearly, we plotted the values in a scatterplot with MTD in the x-axis and $R_{a}$ in the y-axis (Figure 20). The Pearson correlation coefficient of both measurement series leads to 0.9681, which indicates a strong linear relationship.

### 7.3. Area- vs. Line-Based Estimation of the Roughness

To demonstrate the necessity of an area-based roughness determination, we conducted further examinations of the concrete specimens. In particular, we chose a specimen appearing to be very irregular and coarse to carry out further analyses.

Calculating, e.g., the roughness parameter $R_{a}$ for that specimen in the area-based way leads to 0.905 mm. For comparison with the line-wise calculation, we extracted 11 equally spaced lines both in the horizontal and vertical directions from the same point cloud. The extracted lines have a spacing of 4 cm. Calculating the line-based roughness parameter leads to the following measurement series:

The individual values vary substantially as shown in Figure 21. For the horizontal lines, the arithmetic mean is ${\bar{x}}_{h}=0.889\;\mathrm{mm}$ and the standard deviation is $s_{h}=0.249\;\mathrm{mm}$. For the vertical lines, the mean is ${\bar{x}}_{v}=0.891\;\mathrm{mm}$ and the standard deviation is $s_{v}=0.199\;\mathrm{mm}$. Setting up a 95% confidence interval for the horizonal lines results in
(13)$$P\left\{0.722\leq {µ}_{h}\leq 1.056\right\}=95\%\;$$
and for the lines in the vertical direction, it results in
(14)$$P\left\{0.757\leq {µ}_{v}\leq 1.025\right\}=95\%.$$

A normally distributed measurement series of the lines would lead to 5% of the values falling outside the 95% confidence interval (meaning either no line or just 1 line should fall outside the interval). However, in our case, six of the horizontal lines and four of the vertical lines do not fall into the calculated interval for the expected mean of the measurement series. Accordingly, the assumption that the individual profile lines are only subjected to normally distributed noise does not apply. Thus, the deviations of the lines can no longer be justified just stochastically, and this brings us to the conclusion that a single line is unsuitable to specify the roughness of an entire surface. An area-based determination, on the other hand, enables a reliable estimation of the roughness since all measuring points of the entire surface are concerned in the calculation. 

## 8. Conclusions

### 8.1. Summary

This paper introduces a novel camera-based measurement system, which enables high-resolution analysis of technical surfaces to be performed. As a use-case, we demonstrated the roughness estimation using concrete specimens. However, basically every surface that is in the depth of field of the camera and hence can be captured sharp in the images can be measured with the proposed system. The image matching procedure, though, requires the surface to be non-reflective and to have an irregular and non-repetitive pattern in order to guarantee a unique matching of the image points in the images. However, even these constraints can be addressed by preparation of the object surface. For example, the surface can be sprayed with a very thin layer of paint in order to provide a unique pattern on the surface. Hence, generally any material (e.g., concrete elements, metals) could be measured by the system.

Before using the system for measurement purposes, the interior orientation of the industrial camera had to be calibrated. Self-calibration of the camera based on an object without control points—in our case a (flat) concrete surface—led to inadequate results. or this reason, we designed and manufactured a specific 3D calibration test-field with appropriate three-dimensional point distribution. The re-calibration of the camera using the new test-field significantly reduced the correlations of the interior orientation parameters.

For 3D reconstruction of object surfaces, we developed a two-step image matching pipeline, including SfM and DIM. The SfM algorithm is used for the estimation of the exterior orientation of the camera and is implemented using the open-source library OpenCV. For DIM, we utilize the SGM algorithm. Using the programming model of CUDA, we implemented SGM for GPUs in order to minimize the runtime and meet real-time requirements. As a result, the GPU implementation is 47.5 times faster than the pure CPU implementation.

To obtain initial results for the estimation of roughness, we adapted the roughness parameter $R_{a}$ to 3D point clouds and investigated a total of 18 concrete specimens with different surface textures. Comparing the values for $R_{a}$ estimated by our measurement system and the reference values for MTD determined by sand patch method shows a strong linear correlation. To show the necessity of an area-based measurement of the object surfaces, we carried out more detailed investigations with a particular specimen. It turned out that single lines are not significant for the representation of an entire surface, which in return confirms the importance of an area-based determination of the roughness.

### 8.2. Outlook

The determination of the parameter MTD using 3D point clouds could be performed in a similar way like the parameter $R_{a}$, with the difference that the reference plane is placed on the uppermost points of the point cloud. In addition, in this case, the distances between the points and the plane have to be summed and divided by the total number of points. This calculation procedure principally corresponds to the determination of MTD as derived by the sand patch method. In practice, though, the implementation of a calculation procedure for MTD holds some difficulties. The point clouds reconstructed by our measurement system usually consist of several million points, some of which, inevitably, are outliers. Therefore, choosing which points to use when defining the MTD reference plane would be difficult.

In the future, however, our measurement system enables novel opportunities for the investigation of technical surfaces. Many roughness parameters are designed for lines and are not suitable for the description of entire surfaces. Hence, besides the adaptation of further roughness parameters (e.g., MTD, $R_{v}$, $R_{p}$, $R_{t}$, Abbott-Firestone curve) to 3D point clouds, we think about designing new roughness parameters, which represent the surface properties in a better way. In particular, for different use-cases, different parameters should be considered. For example, a distinction has to be made between roughness parameters used to estimate the amount of coating material required to cover the surface and roughness in terms of adhesive bond of a surface. In this context, for example, the investigation of the gradients in the point clouds would be feasible.

The roughness of a 3D reconstructed surface essentially depends on the measurement resolution. For a resolution-independent estimation, the fractal dimension should be considered, as, for example, presented in [19].

As a further outlook, additional empirical studies, including comparisons with different methods (e.g., laser triangulation, sand patch method) have to be done in order to validate the camera-based system and present the method applicability. Therefore, as a first step, the measurement system could prove to be a supplementary measurement system to the sand patch method and after getting approval, it could perhaps become an alternative method.

Current high-end smartphones are equipped with an integrated graphics processor (IGP), which also enables parallel computing. In particular, smartphones with the Android operating-system provide through RenderScript [52] a powerful API for implementing parallel algorithms which can further be executed by the IGP. Therefore, it is conceivable that our image-based measurement method, as presented in this article, can be adapted onto mobile systems as smartphones since they provide all necessary components, such as, for example, a high-resolution camera and powerful processing unit. In addition, since nowadays almost everybody owns a—more or less powerful—smartphone no additional hardware would be necessary.

## Figures and Tables

**Figure 1 materials-14-00158-f001:**
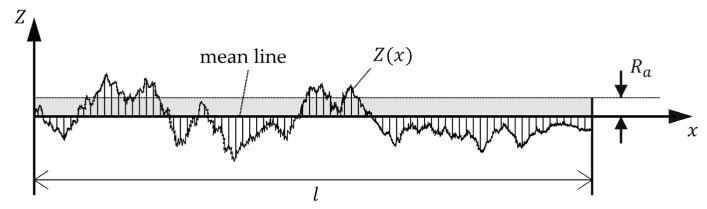
Profile line of a technical surface which is composed of multiple orders of shape deviations.

**Figure 2 materials-14-00158-f002:**
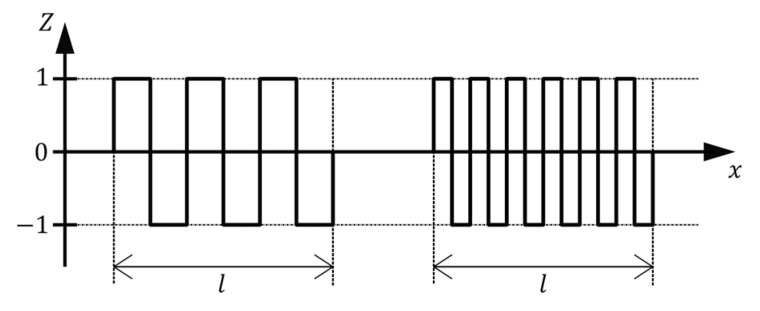
Two lines with different surface characteristics but same result for $R_{a}$.

**Figure 3 materials-14-00158-f003:**
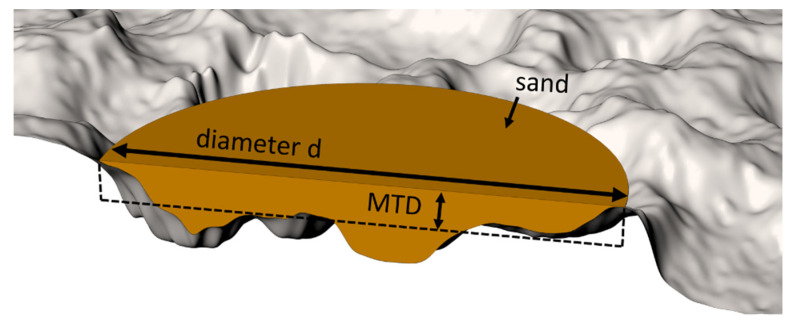
Illustration of the mean texture depth (MTD).

**Figure 4 materials-14-00158-f004:**
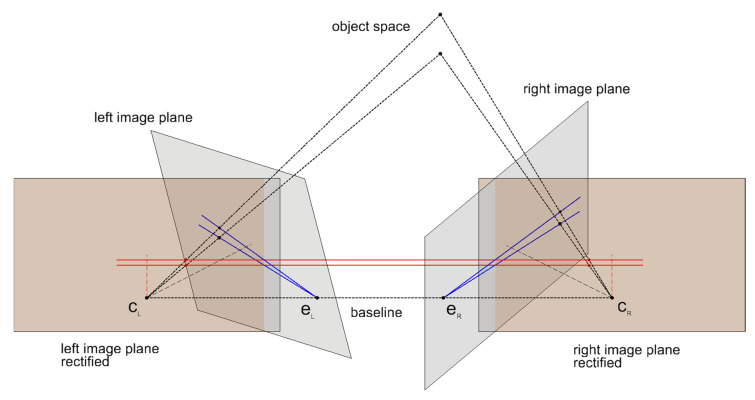
Epipolar geometry—initial image planes (gray) and stereo rectified images planes (orange).

**Figure 5 materials-14-00158-f005:**
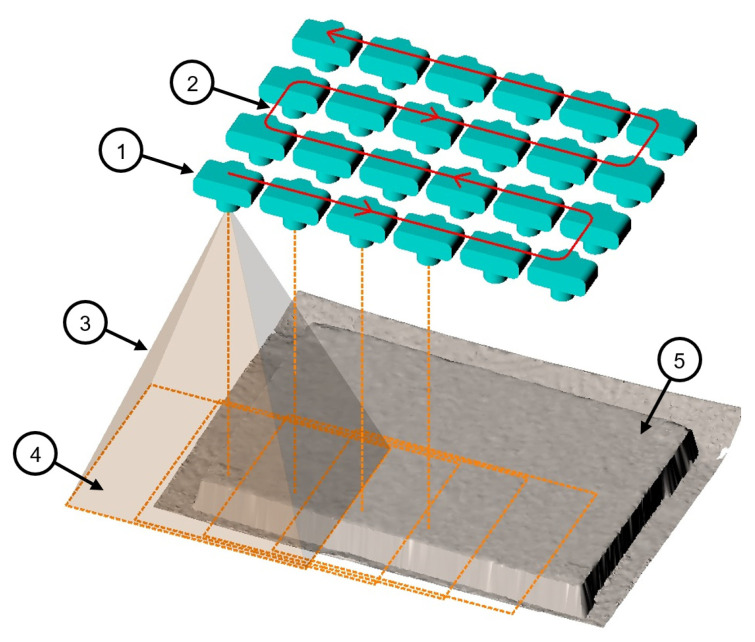
Concept of the recording geometry. 1—camera position; 2—camera trajectory; 3—field of view pyramid; 4—imaging area; 5—concrete surface.

**Figure 6 materials-14-00158-f006:**
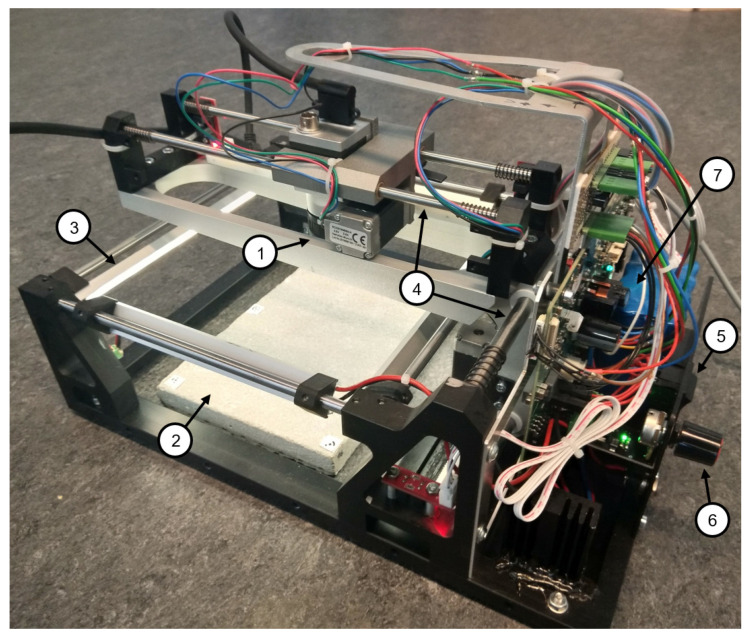
Camera-based measurement system. 1—Industrial camera; 2—concrete specimen; 3—one of the four LED strips; 4—both moving axes; 5—power switch; 6—rotary switch for illumination adjustment; 7—rechargeable battery.

**Figure 7 materials-14-00158-f007:**
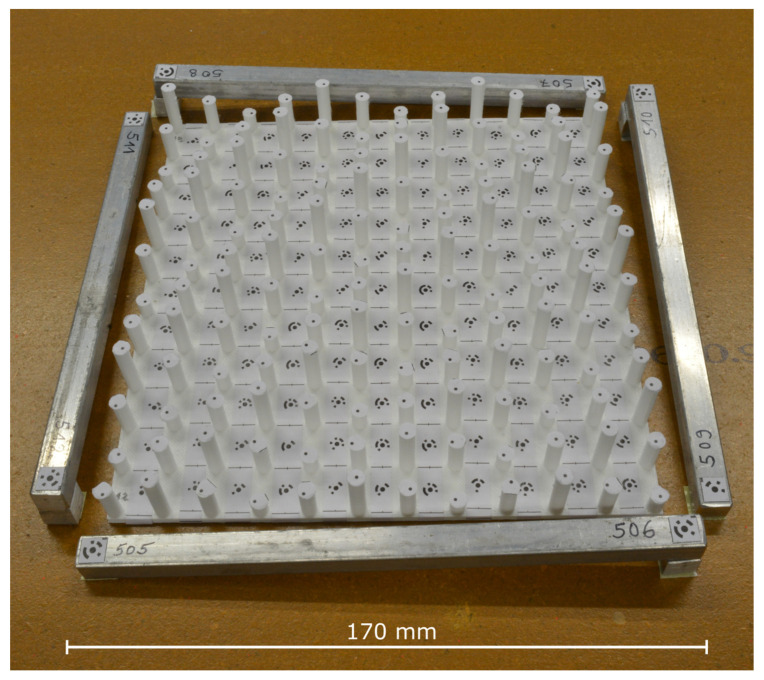
Custom-built 3D calibration test-field.

**Figure 8 materials-14-00158-f008:**
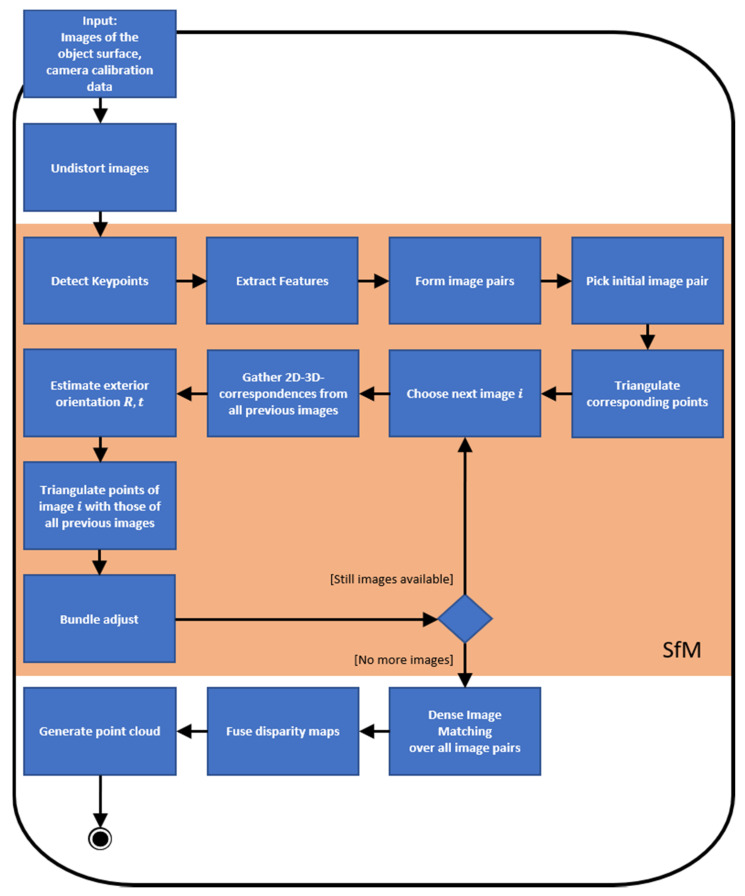
The incremental 3D reconstruction pipeline.

**Figure 9 materials-14-00158-f009:**
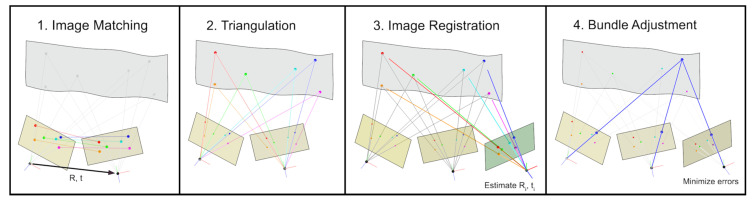
Visualization of the structure from motion procedure.

**Figure 10 materials-14-00158-f010:**
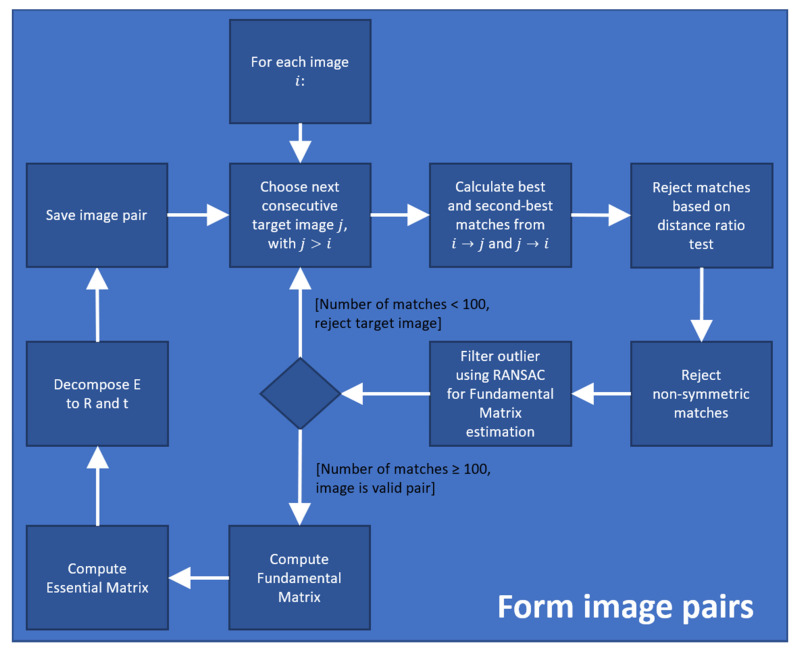
Procedure for image pair formation.

**Figure 11 materials-14-00158-f011:**
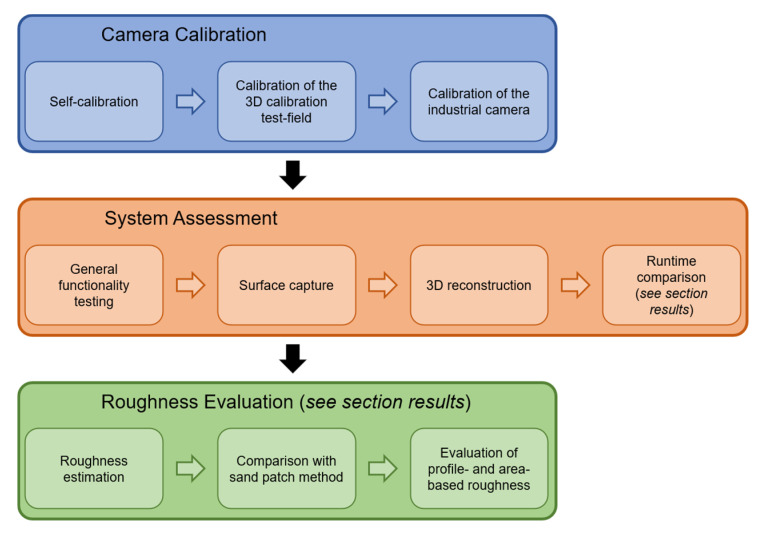
Overview of the conducted experiments.

**Figure 12 materials-14-00158-f012:**
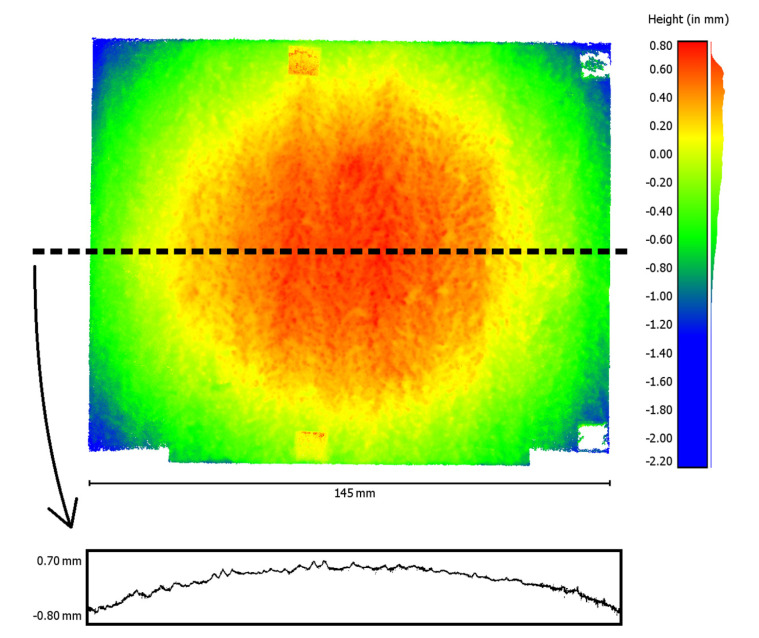
Reconstructed point cloud with conspicuous curvature after self-calibration (**top**, **left**), extracted profile line (**bottom**, 10-fold scaled in height), and histogram of the height values (**right**).

**Figure 13 materials-14-00158-f013:**
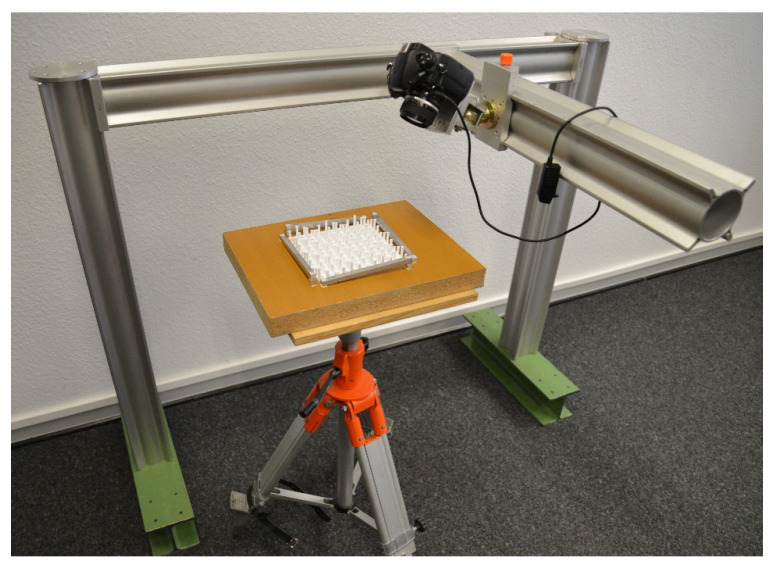
Measurement setup for the calibration of the 3D calibration test-field.

**Figure 14 materials-14-00158-f014:**
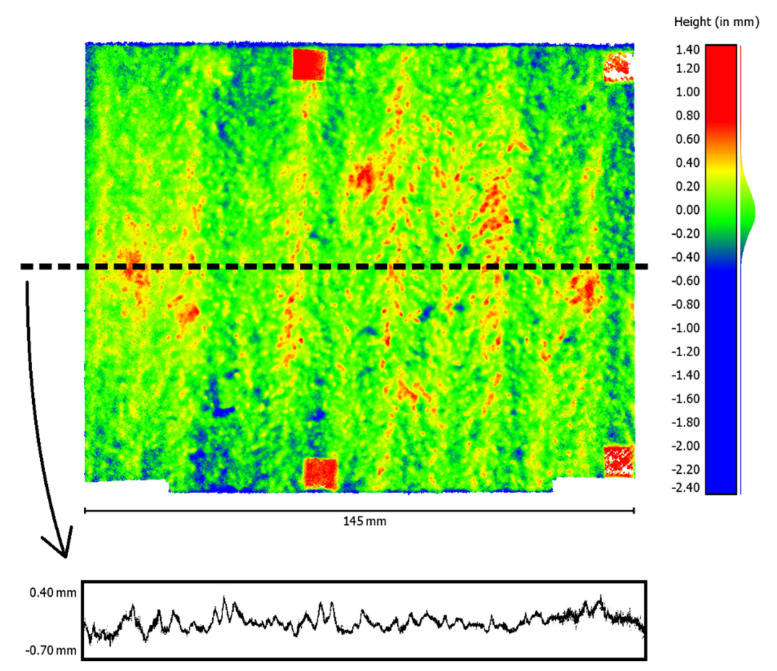
Point cloud (**top**, **left**), extracted profile line (**bottom**, 10-fold scaled in height), and histogram of the height values (**right**) after reconstruction with the new parameter set.

**Figure 15 materials-14-00158-f015:**
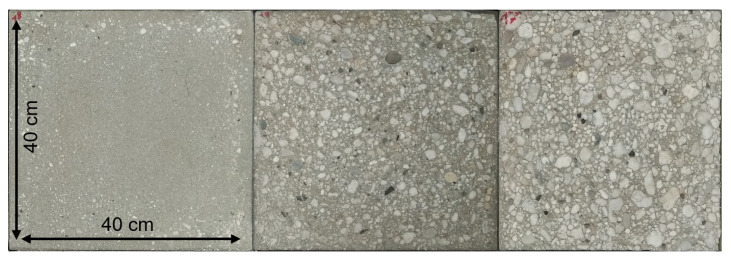
Selection of three concrete specimens with different roughness of the investigated 18.

**Figure 16 materials-14-00158-f016:**
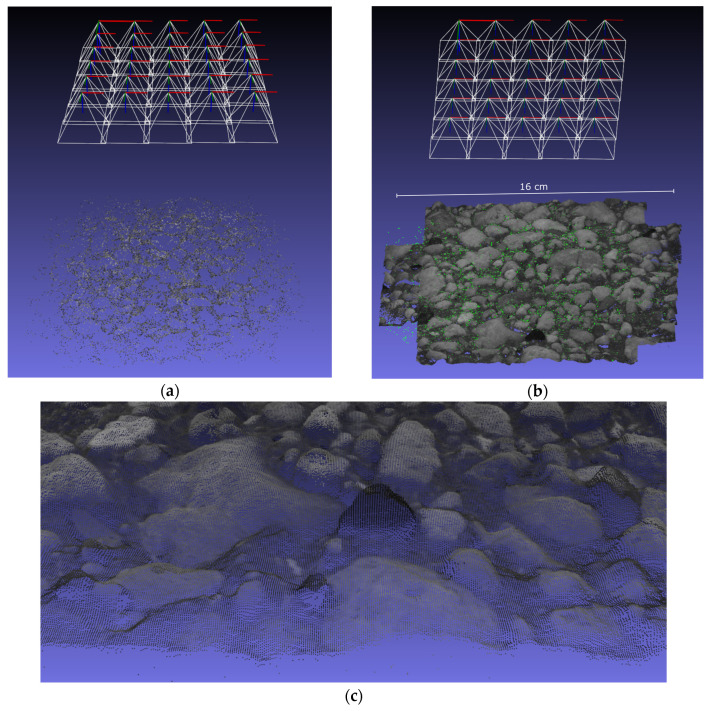
3D point cloud reconstruction of a particular concrete specimen: (**a**) Reconstructed camera poses including the sparse point cloud after SfM procedure, (**b**) generated dense point cloud after Dense Image Matching (DIM), and (**c**) zoom-in of the dense point cloud.

**Figure 17 materials-14-00158-f017:**
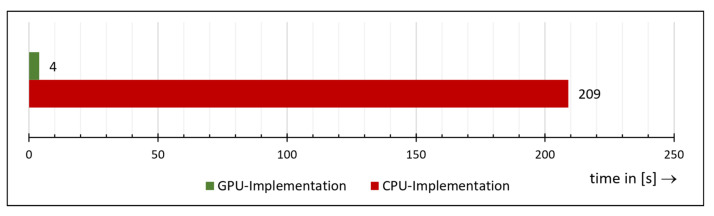
Comparison of the total runtime between the pure CPU implementation and the Graphics Processing Unit (GPU)-accelerated implementation.

**Figure 18 materials-14-00158-f018:**
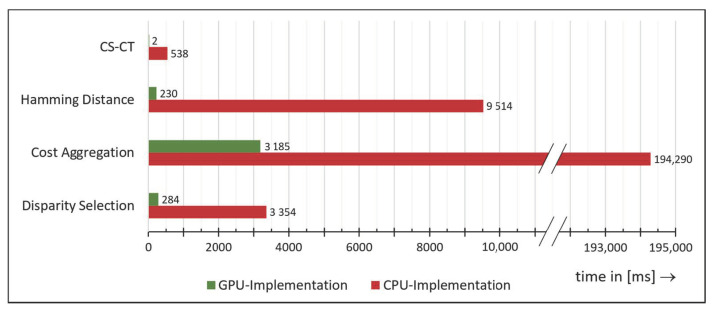
Runtime comparison of the single algorithms of the Semi-Global Matching (SGM) pipeline.

**Figure 19 materials-14-00158-f019:**
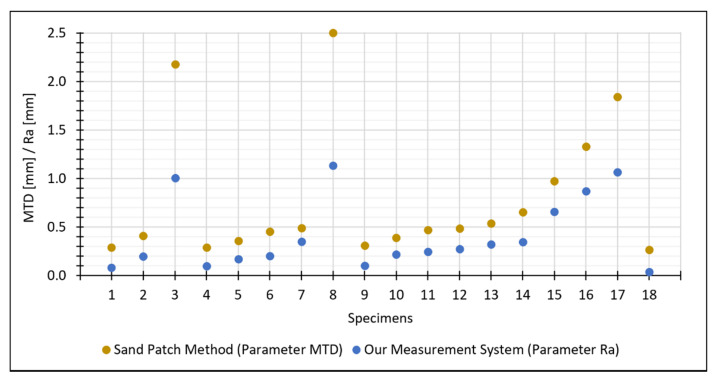
Comparison between $R_{a}$ estimated by our measurement system and the references values for MTD determined by the sand patch method.

**Figure 20 materials-14-00158-f020:**
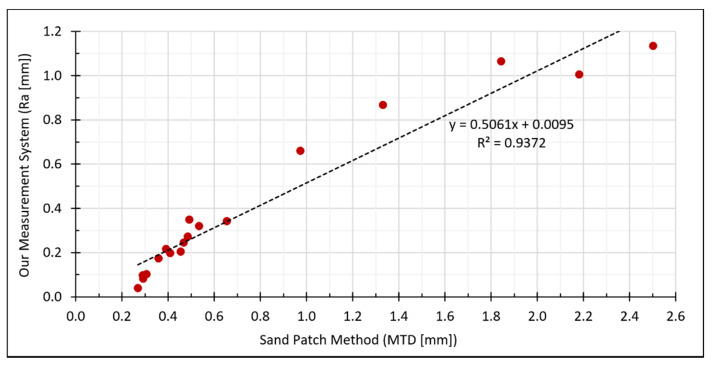
Correlations between $R_{a}$ estimated by our measurement system and the reference values for MTD determined by the sand patch method.

**Figure 21 materials-14-00158-f021:**
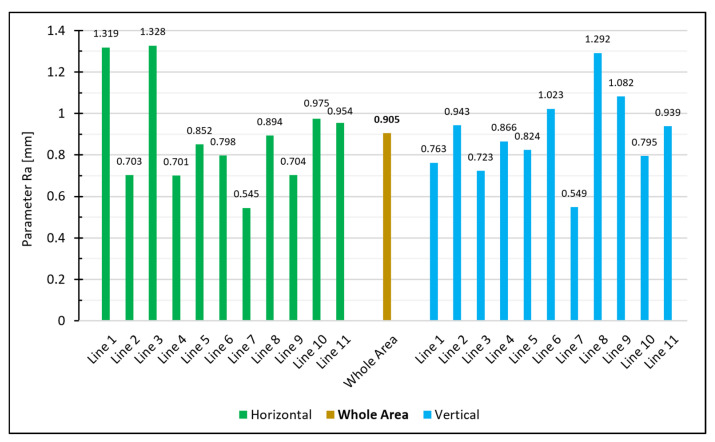
Calculated $R_{a}$ for the extracted lines and for the whole area.

**Table 1 materials-14-00158-t001:** Shape deviations of technical surfaces (according to the German standard DIN 4760).

Shape Deviations		
1. Order Form deviation	Curvature, Unevenness	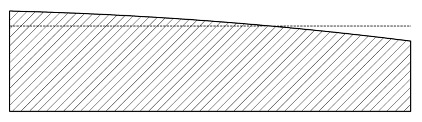
2. Order Waviness	Waves	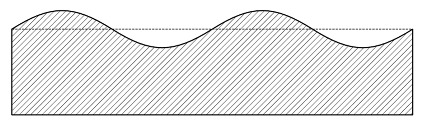
3. Order Roughness	Grooves	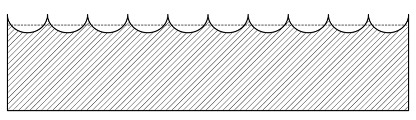
4. Order Roughness	Ridges, Scales, Crests	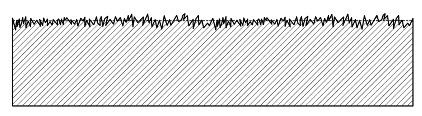
5. Order Roughness	Microstructure of the material	not easily presentable in image form
6. Order	Lattice structure of the material	not easily presentable in image form

**Table 2 materials-14-00158-t002:** Specifications of the used industrial camera Basler acA3800-14um.

Specification	Value
Resolution (H × V)	3840 pixel × 2748 pixel
Pixel size (H × V)	1.67 µm × 1.67 µm
Bit depth	12 bits
Signal-to-noise ratio	32.9 dB
Mono/Colour	Mono
Shutter technology	Rolling shutter

**Table 3 materials-14-00158-t003:** Dependencies of the parameters of interior orientation after self-calibration.

	f	cx	cy	k1	k2	k3	p1	p2
f	1.00	−0.06	0.15	−1.00	0.99	−0.97	0.22	0.13
cx		1.00	0.05	0.06	−0.06	0.06	0.04	0.01
cy			1.00	−0.15	0.14	−0.13	0.04	−0.02
k1				1.00	−0.99	0.97	−0.22	−0.13
k2					1.00	−0.99	0.22	0.13
k3						1.00	−0.21	−0.13
p1							1.00	0.03
p2								1.00

**Table 4 materials-14-00158-t004:** Parameters of the interior orientation after calibration using the 3D calibration test-field.

Parameter		Value	Std. Dev.
f		8.2545 mm	0.0007 mm
cx		0.0737 mm	0.0012 mm
cy		0.0051 mm	0.0007 mm
k1	$\left(\cdot 10^{-4}\right)$	−43.8231	0.2449
k2	$\left(\cdot 10^{-7}\right)$	565.8091	36.7678
k3	$\left(\cdot 10^{-10}\right)$	−10,555.4693	1665.3598
p1	$\left(\cdot 10^{-5}\right)$	5.1708	0.2624
p2	$\left(\cdot 10^{-5}\right)$	10.2455	0.2549

**Table 5 materials-14-00158-t005:** Correlation coefficients of the parameters of interior orientation after calibration using the 3D calibration test-field.

	f	cx	cy	k1	k2	k3	p1	p2
f	1.00	0.00	0.01	0.03	−0.01	0.00	0.00	0.00
cx		1.00	0.00	−0.01	0.01	−0.01	0.72	0.00
cy			1.00	0.00	0.00	0.00	0.00	0.38
k1				1.00	−0.98	0.93	0.01	0.00
k2					1.00	−0.99	−0.01	0.00
k3						1.00	0.01	0.00
p1							1.00	0.00
p2								1.00

## Data Availability

“Not applicable” for studies not involving humans.

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
