# Peer review of "A Novel Camera-Based Measurement System for Roughness Determination of Concrete Surfaces"

_materials, 2020, doi:10.3390/ma14010158_

Round 1
Reviewer 1 Report
Please consider the following comments while improving the paper:
1. Although I appreciate the effort made in the system development its superiority and advantages should be clearly stated.
2. Please clearly state what is the novelty of your research because area-based surface reconstruction is possible with other methods.
3. It is difficult to follow the system described in Section 5.1.2. If possible please use more diagrams/figures.
4. Algorithms should be strictly described or referenced.
5. It is essential to compare the results of the presented method with other non-contact methods (not only sand-patch). That will present the method applicability. Reference given to paper [9] is not satisfactory.
6. Please consider if lines 79-88 are necessary.
7. Section 3.1.2. Please mention that other roughness parameters exist and where one can find them.
8. Section 8.1 repeats paper content. Is it necessary as a conclusion?
Reviewer 2 Report
The presented investigation contains a very interesting topic. The research covers the issues of roughness assessment of building materials with the use of an innovative measuring system based on the industrial camera. The article has a proper structure and is written in professional, methodically correct language. The part concerning the introduction, theoretical approach, description of the measurement system or methodology does not raise any objections. Minor notes are attached to the text.
However, the description of the experiment should be significantly changed and improved . Authors should considered the following issues:
(1) At the beginning, there is no general description of the stages involved. Authors should preper the overview of the performed experiments and presented i.e. as a graph How was the experiment conducted?
(2)The self-calibration text is an obvious confirmation of the photogrammetric solutions used. On a flat object and with such a geometry of images, the results could not be different.
(3)The figures are illustrative without color legend or scale. It is clear from the camera calibration that the Z coordinate is determined with less accuracy. The reasons are indicated, but has any attempt been made to improve it? Why is such accuracy assumed to be good?
(4) How was the sample measurement carried out? How were the scale bars used? Were they treated as warp points and were there control points in the alignment stage? There is no description of the results from reports from intermediate stages of calculations.
(5) What was the reprojection error, at what level of the pyramid the detector was used, etc. This part of the article lacks an extended assessment of the quality of the results obtained at certain stages.
(6) How was the minimization of the indentation implemented, in how many images?
(7) Finally, the visualization of the obtained point clouds on the tested samples was not shown, but only the camera positions. There is no more information about the difference in surface quality of the tested samples? Was the roughness similar for everyone? How did the shorts differ between the 18 samples. Of course, the 3D visualization of the generated point clouds can be selected and exemplary studies presented. When analyzing the results, the interval information for vertical and horizontal profiles is missing. How they were spread over the samples There is only a statement that there were 11 of them.
Generally, this part needs to be re-edited so that it is possible to evaluate not only the measurement itself, but also individual stages of the calculation process and the quality of the obtained results.

Reviewer 3 Report
The article: A Novel Camera-based Measurement System for Roughness Determination of Technical Surfaces present a new and interesting method for surface roughness analyze with many potential applications.
Few comments that the authors should pay attention to:
- the abstract and introduction are too long - please narrow the information to your new results reported to literature
- theoretical background is like a students class course and must be limmited only to specific information really important for the article, and removed figures 1-4. Any way the figures (if you decide to keep them) needs references.
- figure 6: mention which figure is a) and which b) - and provide a scale bar for b)
- for figures 12 and 13 also provide a bar scale (m, cm or mm)
Reviewer 4 Report
This is a well-written article on determining the roughness of technical surfaces especially construction-related. A piece of new measurement equipment Is developed to capture the roughness. The measurement system, software, working methodology, and resulting roughness estimates are presented. Important details on practical use seem to be missing though
Some comments for the authors to consider
- Abstract: The results should be quantified (positive and negative). How does your system compare with the conventional machine measurement results?
- Introduction: A very well instruction and related work. May be too long.
- Section 3.1.2: The authors provide a reference to the correct standard for surface texture measurement. However, use the wrong terminologies. “Roughness” is not a correct definition. Surface textures are more scientific.
- Ra — arithmetic mean deviation of the assessed profile, not arithmetic mean roughness. Suggest using the same definitions in ISO 4287 and ISO 4288. Revise Rq and MTD as well. The authors use different terms throughout the paper. Line 540.
- Section 4.1: 60-80 % overlap is very high. How much time is required to capture a particular length and stitch? What are the reasons for this overlap? Advantages and limitations of reducing the overlap?
- Suggest to include labels in Fig 5 and Fig 6
- Fig 11: Please provide the values for the height scale in the image. Both top and bottom with units
- The authors use 4 decimals to represent results. Is this significant? Since the unit is in mm, I would recommend reducing to 2 decimal points.
- Major comment: The authors initially say that Ra is not a correct representation, as two different surfaces can have the same value. This is correct. But throughout the paper, the authors emphasize Ra. How about other parameters in ISO 4287? Rz is a very good representation of surface heights. I would probably like to see how your equipment captures Peak to Valley distance of surfaces.
- Major comment: Inorder for others to accept your equipment, you need to compare the results from your machine with conventional machines. I like to probably see a comparison of the results with a standard commercial machine. Especially Ra, not something different. What is the deviation? Is it high or low? That will provide good confidence for the readers.
- Finally, would it be possible to measure metals using this machine? What is the lowest scale of roughness that can be detected using this machine? Can it perform micro scale measurements? The authors can include details by measuring fine concrete surfaces as well.
- Recommendation: I like to suggest the authors consider artefacts or surface roughness gages with different roughness levels and test the machine extensively at various scales to gain confidence. This could be future work. This is because in your Fig 17, most results are biased towards 0.2-0.8 mm, only 5 point lie > 1.0? What is your justification for this? If the data points from 0.2-0.8 is removed what is the R-sq?
Round 2
Reviewer 1 Report
- Unfortunately, the explanation given on the research novelty is not satisfactory. Area-based, contactless, digital surface reconstruction is possible with other low-cost, lightweight methods.
- It is essential to compare the results of the presented method with other non-contact methods (not only sand-patch). That will present the method applicability.
Reviewer 2 Report
The submitted article after proofreading was assessed with comments in the cover letter.
Undoubtedly, it describes a new solution for the use of an industrial camera and a measuring system in roughness testing of building materials. The authors of the article responded to the comments and corrected some parts of the text. Not all of them have been included. However, when assessing the overall text of the article, it can be assumed that this form may already be published. The article is written in professional, methodically correct language. The part concerning the introduction, theoretical approach, description of the measurement system or methodology does not raise any objections.
However, there are a few corrections that can be made.
The adopted arrangement of chapters and subchapters is unusual. E.g:
- Theoretical background,
3.1 Defining roughness
3.1.1 Shape deviations
It is a strange arrangement of the article to present successive breakdowns of a chapter without any introduction or comment in each. In each of these chapters there should be at least one sentence related to it. In this form, according to the reviewer, it is an artificial division.
The caption under Figure 16 should also be changed. The captions used (top, left) and (top, right) are rather unprofessional. Symbols should be used, e.g. (a), (b), (c).
Reviewer 4 Report
I disagree with the author's answers to comment 10. The authors should show how their system compares with conventional machines (since you mention that you have already started comparing). Or at least should provide the comparison for reviewer satisfaction if they do not wish to include it in the manuscript.
Labels are still missing in Fig.5 and Fig.6
